# Effects of Processing Conditions and Plasticizing-Reinforcing Modification on the Crystallization and Physical Properties of PLA Films

**DOI:** 10.3390/membranes11080640

**Published:** 2021-08-20

**Authors:** Shuo Wang, Baodong Liu, Yingying Qin, Hongge Guo

**Affiliations:** School of Light Industry Science and Engineering, Qilu University of Technology, Jinan 250353, China; 10431200886@stu.qlu.edu.cn (S.W.); lbd980918@163.com (B.L.); 17862977017@139.com (Y.Q.)

**Keywords:** polylactic acid blown film, processing conditions, crystallization, physical properties, DSC analysis

## Abstract

The polylactic acid (PLA) resin Ingeo 4032D was selected as the research object. Epoxy soybean oil (ESO) and zeolite (3A molecular sieve) were used as plasticizer and reinforcing filler, respectively, for PLA blend modification. The mixture was granulated in an extruder and then blown to obtain films under different conditions to determine the optimum processing temperatures and screw rotation. Then, the thermal behaviour, crystallinity, optical transparency, micro phase structure and physical properties of the film were investigated. The results showed that with increasing zeolite content, the crystallization behaviour of PLA changed, and the haze of the film increased from 5% to 40% compared to the pure PLA film. Zeolite and ESO dispersed in the PLA matrix played a role in toughening and strengthening. The PLA/8 wt% zeolite/3 wt% ESO film had the highest longitudinal tensile strength at 77 MPa. The PLA/2 wt% zeolite/3 wt% ESO film had the highest longitudinal elongation at 13%. The physical properties depended heavily on the dispersion of zeolite and ESO in the matrix.

## 1. Introduction

The global annual consumption of plastics reached 500 million tons in 2019; the huge amount of discarded plastic caused severe and irreversible environmental pollution. The development of biodegradable plastics can fundamentally solve the problem, and has therefore become one of the most important research topics [1].

Polylactic acid (PLA) is a new biobased and renewable biodegradable material. It is made of materials containing starch from renewable plants (such as corn, wheat, and cassava) [2,3,4]. Glucose is obtained from starch raw material through saccharification, and then high-purity lactic acid is produced by fermentation of glucose and certain bacteria. Then, PLA with a certain molecular weight is synthesized by chemical synthesis. It can be completely degraded by microorganisms in nature under specific conditions, resulting in carbon dioxide and water, and is recognized as an environmentally-friendly material [5,6,7,8]. Some brands of PLA resin with a strength up to 54.69 MPa are accepted as representatives to replace petroleum-based plastics, such as PE and PP. However, the inherent high brittleness and low toughness of PLA (e.g., elongation at a break of 7.69%, notched impact strength of 2.33 kJ/m^2^) and its high cost severely restrict its wide application [1]. Used as packaging film, PLA resin has high crystal transparency, good solvent resistance (insoluble in alcohols, fats, hydrocarbons, edible oils, and mechanical oils), lower temperature heat sealing ability, better printability than polyolefins, and good ink retention, which can retain the flavour and package aesthetics of foods to a greater extent [9]. However, it still has low flexibility, slow crystallization rate, low crystallinity and weak barrier properties [10].

Recently, many experts have paid attention to the modification of PLA by physical and chemical methods to overcome its inherent shortcomings [11,12,13,14,15,16,17,18,19]. PLA resin blended with cellulose [20], cellulose fiber [21,22] and poly (butylene adipate-coterephthalate) (PBAT) will also improve the packaging properties of PLA film [23]. Fillers, such as montmorillonite (MMT), are natural clays with a high specific surface area that can potentially produce “nano barrier walls” within the nanocomposites. Organically modified montmorillonite (OMMT) is used to achieve an improved dispersion effect within the PLA matrix [24]. Jong et al. [25] fabricated PLA-based composite films with different types of nanoclays by solvent casting. They found that the water resistance of PLA films improved to different degrees.

A biodegradable citrate ester plasticizer and vegetable oil-based epoxidized soybean oil (ESO) plasticizer toughened PLA in nearly ten years. The mechanical properties, thermal resistance, crystalline behaviors and processing performance changed [1,26]. Copolymerization with other monomers is another approach that has been used to improve the properties such as stiffness, permeability, crystallinity, and thermal stability [8,27,28,29]. Covalent immobilization of polypeptides on PLA films can make it an active packaging [30]. Adding filler and plasticization was explored as an economically viable way to modify the properties of PLA resin. Ingeo Series (Nature work company, Blair, NE, USA) is a high heat-resistant brand in PLA resin and can be used as a basic resin to further optimize the comprehensive properties of modified materials.

Zeolites are hydrous aluminosilicate minerals with network structures. 3A molecular sieve zeolites are layered inorganic nanomaterials that can be used as fillers, nucleating agents and ethanol and water separation in polymers. Zeolites can improve the mechanical properties, increase crystallinity and transfer water vapor and breathing gas as an active packaging bag.

Epoxidized soybean oil (ESO) is recognized as a potential reactive plasticizer with the advantages of low cost, biodegradability, renewability, and availability. ESO is a food-grade additive synthesized from soybean oil through the epoxidation of C=C bonds in the fatty acid chains. ESO is composed of three long and flexible aliphatic chains, and its epoxy functional groups on the chains can react with the –OH and –COOH groups on other polymers. When the long flexible aliphatic chains of ESO were added to polymers, the copolymer resin formed between ESO and polymers had a lower cross-link density, increasing the flexibility of the polymers and significantly reducing the brittleness [31].

Therefore, ESO was used as a plasticizer and zeolite was used as reinforcing fillers, and the PLA blend was mixed by melting extrusion, cooled with water and granulated. Then, the film was blow-moulded to fit a specific blow-up and draw ratio. The studies were focused on different ratios of zeolite and ESO and their interactions with the crystallization and physical properties of PLA films.

## 2. Experimental

### 2.1. Materials

PLA (Ingeo 4032D) was purchased from Nature Works Company (Blair, NE, USA). At a temperature of 190 °C, loading pressure of 2.16 kg and cutting time interval of 30 s, the MFR of Ingeo 4032D PLA was 5.872 g/10 min.

Epoxidized soybean oil (ESO) was purchased from Xinjinlong Plastic Additives Co., Ltd. (Guangzhou, China).

Zeolite (3A molecular sieve, with an average particle size of 2–5 μm) was purchased from Yuanli Chemical Co., Ltd. (Tianjin, China).

### 2.2. Manufacturing Methods

Pretreatment: PLA and zeolite were dried separately in a vacuum drying oven at 80 °C and 105 °C for 4 h.

Mixed granulation: Zeolite was soaked in ESO and stirred with PLA particles in a mixer in different proportions and left in the sealed mixer for 24 h (recipe as per Table 1), and the granular resin for the blowing film was produced by the single screw extruder.

Film sample preparation: The die of the extruder was changed from granular to blowing film; the processing temperatures of the four-heating zoom were 155 °C, 170 °C, 175 °C, and 185 °C, the screw rotating speed was 35 rpm, the blow-up ratio was 2.6 and the draw ratio was 5.0. The film was blown to a thickness of between 30 μm and 80 μm.

### 2.3. Performance Testing and Characterization Methods

#### 2.3.1. Analysis of Differential Scanning Calorimetry (DSC)

A differential scanning calorimeter (DSC 1/700) made by Mettler Toledo Co., Ltd (Zurich, Switzerland) of Switzerland was used to measure the isothermal crystallization behaviour and melting characteristics of the PLA film.

An 8–10 mg sample was weighed and sealed in an aluminium crucible disc, and nitrogen (50 mL/min) was used as the protective gas. The initial temperature was 20 °C, the temperature was raised to 200 °C at a heating rate of 10 °C/min [22], and the temperature was kept constant for 3 min to eliminate the thermal history. The sample was then cooled to room temperature and maintained at a constant temperature for 3 min. Then, the temperature was again raised to 200 °C at 10 °C/min, and the second heating curve was recorded. The crystallinity (*X_c_*) was calculated using the following equation:(1)$$X_{c}=\frac{\Delta H_{m}}{\Delta H_{m}^{0}\times w}\times 100\%$$
where Δ*H_m_* stands for the measured melting enthalpy of the blends; $\Delta H_{m}^{0}$ is the melting enthalpy of PLA with 100% crystallinity in the ideal state, which is 93.7 J/g; and *W* is the mass fraction of PLA in the mixture.

#### 2.3.2. Analysis of Polarizing Microscopy

The thin plate sample was pressed between two cover glasses (20 mm × 20 mm). The sample was heated at 200 °C for five minutes and then cooled naturally to room temperature. Finally, a polarizing microscope (NP-800RF/TRF) made by Henan Xiongdi Instrument Equipment Company of China was used to observe the crystal form.

#### 2.3.3. Transparency Test

According to the GB/T 2410-2008 test method for the transmittance and haze of transparent plastics, a transmittance/haze tester (WGT-S) made by Labthink Electromechanical Technology Company of China was used to measure the transmittance and haze of the PLA film. Equations (2) and (3) below represent the transmittance and haze, respectively, where *φ*_1_ is the luminous flux incident on the surface of the material, *φ*_2_ is the luminous flux passing through the material and *φ*_d_ is the scattered luminous flux.
(2)$$Tt=\frac{\varphi_{1}}{\varphi_{2}}\times 100\%$$
(3)$$H=\frac{\varphi_{d}}{\varphi_{2}}\times 100\%$$

#### 2.3.4. Analysis of Scanning Electron Microscopy (SEM)

The sample was cut from the film into a small block according to the requirements of the instrument. The impact section (film surface) of the sample was sprayed with gold, and a scanning electron microscope (Regulus 8100, Tokyo, Japan) made by Hitachi High-tech Company of Japan was used to observe the cross-sectional morphology.

#### 2.3.5. Film Mechanical Properties Test

According to the GB/T 1040.2-2006 and GB/T 16578.1-2008 plastic tensile test methods, an intelligent electronic tensile testing machine (XLW (EC)) made by Labthink Electromechanical Technology Company of China was used to measure the mechanical properties of the PLA blend film. The longitudinal tensile strength test was performed at a tensile speed of 100 mm/min (Figure 1). The longitudinal tear strength test was performed at a tear speed of 200 mm/min (Figure 2). The samples were cut along the blown direction. Each experiment was the average of more than 5 samples.

## 3. Results and Discussion

### 3.1. Thermodynamic Properties and Processing Conditions of Polylactic Acid Resin

The Ingeo 4032D PLA resin has a low melt flow rate and poor fluidity and can therefore form stable film bubbles during blow moulding. When PLA resin is used as a packaging film, it requires good heat resistance, stable performance, and poor degradation. The DSC curve of the PLA resin (Figure 3) showed that the melting temperature T_peak_ was 163.66 °C. There was no obvious degradation when the temperature was raised to 200 °C.

The single screw extruder is divided into four processing stages: the first is the solid conveying zoom, which is responsible for pushing the solid PLA from the hopper forward, and the temperature should be lower than the melting point of PLA to avoid the softening and melting of granular materials at the blanking port, which can cause bridging phenomena and affect the raw materials entering the extruder. The second is the melting zone where the material begins to soften and melt under the friction of the barrel screw and external heating. The third is the melt conveying zone, where the temperature should reach the melting point, causing the material to melt completely, and the sample should be pushed to the die evenly and stably. The fourth is the die from which the material is extruded.

With increasing temperature, the viscosity of the material decreases, and the fluidity increases. Therefore, to prevent melt backflow, the temperature of the four zones should be increased in turn (marked as T1, T2, T3). When the same weight of resin was added, the rotation speed was 35 rpm, and the influence of temperature on the change in extrusion torque was recorded (Figure 4). The torque curve can be marked in several sections.

According to the three curves, the balance torque is T2 > T3 > T1. It is important to choose the right processing temperature. If it is too high, it may cause thermal degradation. If it is too low, the material may not melt completely. T2 was selected for the processing condition because the torque is relatively stable and slightly higher than the others, which promotes the uniform mixing of materials.

### 3.2. Effect of Zeolite Addition on the Crystallinity and Transparency of the PLA Film

Commercial PLA is a blend of PLLA and PDLA or copolymer PDLLA obtained by the polymerization of LLA and DLLA, respectively [8]. Many important properties of PLA are controlled by the ratio of the D- to L-enantiomers used and the sequence of arrangement of the enantiomers in the polymers. PLLA constitutes the main fraction of PLA derived from renewable sources since the majority of lactic acid obtained from biological sources exists as LLA. PLA with a PLLA content higher than 90% tends to be crystalline, while PLA with a lower optical purity is amorphous. The melting temperature (Tm), glass transition temperature (Tg), and crystallinity of PLA decrease with decreasing amounts of PLLA [4,27,28,29]. PLA can be amorphous or semicrystalline depending on its stereochemical structure and thermal history.

Figure 5 presents the DSC second heating curves of PLA and PLA/zeolite blend films at 10 °C/min. The nonisothermal crystallization and melting parameters of PLA and PLA/zeolite composites obtained from DSC curves are summarized in Table 2.

Tg: As seen in Figure 5 and Table 2, the glass transition temperature of the blend is slightly lower than that of pure PLA, attributable to the interaction that occurs when ESO and the PLA molecular chains are added; this interaction replaces the interaction between the original PLA molecular chains, weakening the force between the molecular chain segments and reducing the glass transition temperature.

Tc: Compared with other semicrystalline polymers (polyethylene or polypropylene), PLA crystallizes more slowly. Without further treatment, it is difficult to improve the crystallinity and heat resistance temperature. The appearance of a cold crystallization peak indicates that some of the PLA cannot crystallize due to the supercooling rate from 200 °C to room temperature. After the second heating, the PLA that could not crystallize became crystallized as the temperature drops, and a large amount of heat was released. When 10 wt% zeolite was added, the cold crystallization peak became obviously shorter and wider and migrated to the low-temperature region. The results show that the addition of zeolite can obviously accelerate the crystallization rate of PLA.

Tm: The nucleation of zeolite greatly influences the crystallization process and morphology of PLA, and the melting curves show obvious melting double peaks. One opinion is that the low-temperature peak is due to the melting of the initial crystal, while the high-temperature peak is due to recrystallization that occurs during the heating process, which improves the ordered arrangement of the chain segments in the crystal plate [32]. Another opinion is that PLA can crystallize into three forms (α, β, and γ), a phenomenon generally referred to as polymorphism [33]. Polymorphism in materials science refers to the existence of more than one crystalline structure in a solid material with the same chemical composition [34]. The α-structure, which has a Tm of 185 °C, is more stable than the β-structure, which has a Tm of 175 °C [35]. The β-structure of PLA has also been widely investigated [36,37,38,39,40,41,42] since Eling et al. [38] first detected the presence of the β-structure upon hot drawing of the melt-spun or solution-spun PLLA fibres under a high draw ratio. β-structure crystals are generally prepared by stretching their α-structure at high temperature and a high draw ratio. Therefore, it can be inferred that the blown film has a β-structure.

With increasing zeolite content, the difference between the melting temperature of the α-structure and β-structure crystals and the crystallinity increases, and more stable α-structure crystals appear.

PLA was pressed between two coverslips on a hotplate and heated to 200 °C for melting and then cooled to room temperature for nonisothermal crystallization. Figure 6 shows a polarizing microscope photograph of the blend amplified 100 times. Figure 7 shows the blend amplified 1000 times.

With the increase in zeolite proportion, the number of spherulites observed in the vision field obviously increases. The average diameter of spherulites with different zeolite proportions is calculated in Table 3. The larger the zeolite proportion is, the smaller the average diameter of the spherulites is. This proves that zeolite acts as a nucleating agent.

Figure 8 shows the effect of zeolite on the transmittance and haze of the PLA films. The light transmittance remains stable, but the haze increases considerably. Zeolite as a nucleating agent causes light scattering in small crystals.

### 3.3. Effect of ESO and Zeolite Addition on the Structure and Physical Properties of PLA Films

To maintain the biodegradability and nontoxicity of the PLA, green plasticizers and vegetable oil-based plasticizers are widely used to improve the brittleness and reduce the price of PLA mixture, as the price of ESO is lower than of PLA resin [43]; 3% of ESO can also reduce the cost of product.

ESO was prepared from soybean oil after oxidation treatment. It is colourless and nontoxic, meets the requirements of green environmental protection, and is biodegradable and inexpensive. Its boiling point is 150 °C, it has excellent thermal stability, light stability, water resistance and oil resistance, and it can endow products with higher strength and better weather resistance. ESO contains many epoxy groups (Figure 9). The oxygen-containing ternary ring structure on these groups has a high tension and can polymerize with hydroxyl groups, anhydride groups and unsaturated groups containing active hydrogen atoms.

Plasticizers can improve the brittleness of PLA and reduce the strength and rigidity and other mechanical properties, though filling reinforcers are needed. Zeolite is a hydrous aluminosilicate mineral with a network structure. All kinds of zeolite crystals have nanoscale crystal structures and uniform microporous structures, which gives them high specific surface areas, stable chemical properties, good thermal stability, and high adsorption capacities. The internal channel size is between 3–10 A, which fits the characteristics of adsorbing molecules. The 3A molecular sieve is a zeolite crystal with the structure of two four-membered rings. The average particle size is 2–5 μm. It can be used as a nanomaterial to reinforce the matrix.

PLA is a semicrystalline material in the solid-state (Figure 10). Zeolite was soaked in ESO and mixed with PLA to granulate, and then it was blown into a film. It is thought that both the zeolite and ESO insert into the amorphous phase of PLA to form a blended structure in a reinforced network to improve the rigidity, yield stress and machinability of the material.

Figure 11 shows an SEM image of the dispersion of zeolite particles on the surface of the PLA matrix after it was fixed with 3 wt% ESO and the mass fraction of zeolite was increased. The principle of SEM is based on the different conductivity of materials. The conductivity of zeolite is much higher than that of PLA. So the highlighted particles must be zeolite and the dark continuous phase is PLA resin [24]. While the image of pure PLA shows a dark surface, this image contains white dots that are believed to be zeolite particles. The zeolite particles can be evenly dispersed in the PLA matrix at a 2 wt% dose. Figure 12 shows the inferred average size of the particle. Particles between 1 and 100 nm are considered to be nano-reinforced [44]. At doses greater than 2 wt%, most of the particles are oversized. Agglomeration occurs, which may greatly compromise the physical properties.

Tensile samples were cut along the blown direction of the film. The longitudinal tensile strength and elongation are shown in Figure 13.

This figure shows the effect of the zeolite mass fraction on the physical properties of PLA films. The tensile strength curve has a peak at 8 wt% zeolite, and the elongation curve has a peak at 2 wt% zeolite. The sample with 10 wt% zeolite has a higher tensile strength and higher elongation than pure PLA. It seems that adding 10 wt% zeolite and 3 wt% ESO can even improve the comprehensive properties of PLA, but if the dispersion is finer, then the performance is better.

## 4. Conclusions

A melting extrusion blow moulding method was used to process a series of PLA/ESO/zeolite blend films under the most ideal experimental conditions in our laboratory. Zeolite acted as a nucleating agent, increasing the crystallinity and haze of the PLA film, decreasing the size, and changing more crystal forms from β-structures to α-structures. SEM indicated that the zeolite particles agglomerated when the mass fraction of zeolite was higher. When 10 wt% zeolite and 3 wt% ESO were added, the tensile strength and elongation were also higher than those of pure PLA. Zeolite was also shown to be a reinforcing filler and ESO was shown to be a plastizer. Our future work will be focused on the dispersion of zeolite to gain an economically active packaging film.

## Figures and Tables

**Figure 1 membranes-11-00640-f001:**
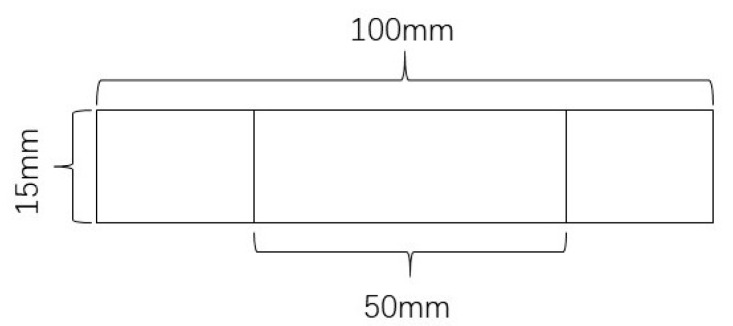
The size of the sample for the tensile strength test.

**Figure 2 membranes-11-00640-f002:**
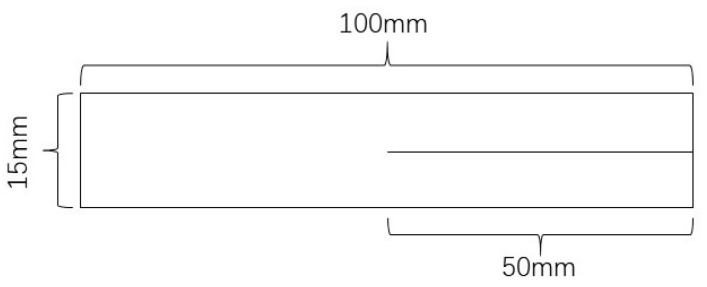
The size of the sample for the tear strength test.

**Figure 3 membranes-11-00640-f003:**
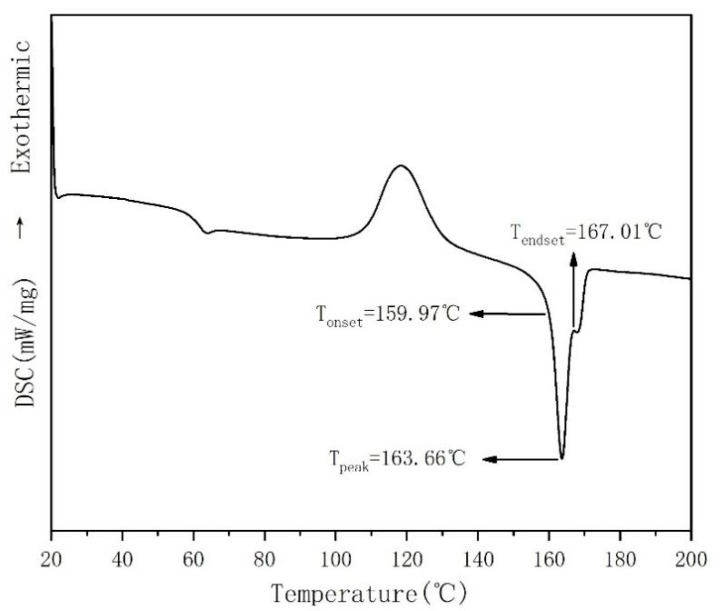
DSC second heating curve of PLA.

**Figure 4 membranes-11-00640-f004:**
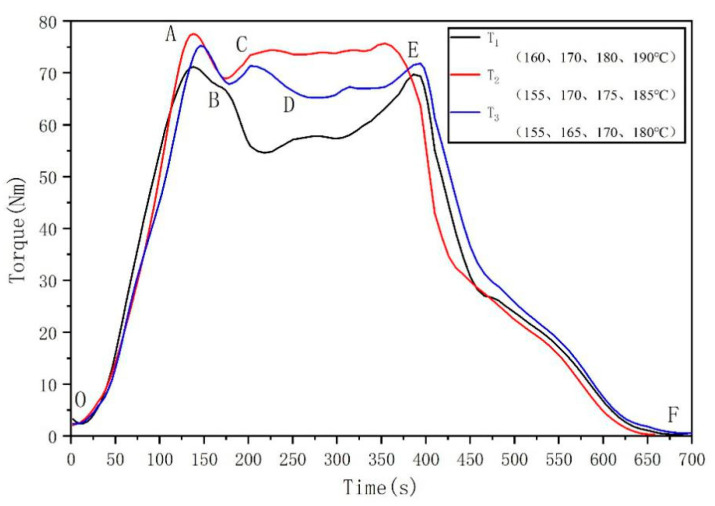
The influence of temperature on the extrusion torque change. OA section: When solid material is added, the torque increases rapidly. AB section: The material is compressed, and the torque drops to point B. BC section: The material begins to soften or melt, and the torque rises to point C. CD section: The material continuously and completely melts under the action of heat and shear, the fluidity is enhanced, the viscosity is reduced, the force of screw on the material is reduced, and the torque is reduced to point D. DE segment: The torque tends to be stable and is maintained near the E point. EF section: With the decrease in material, the torque gradually drops to zero.

**Figure 5 membranes-11-00640-f005:**
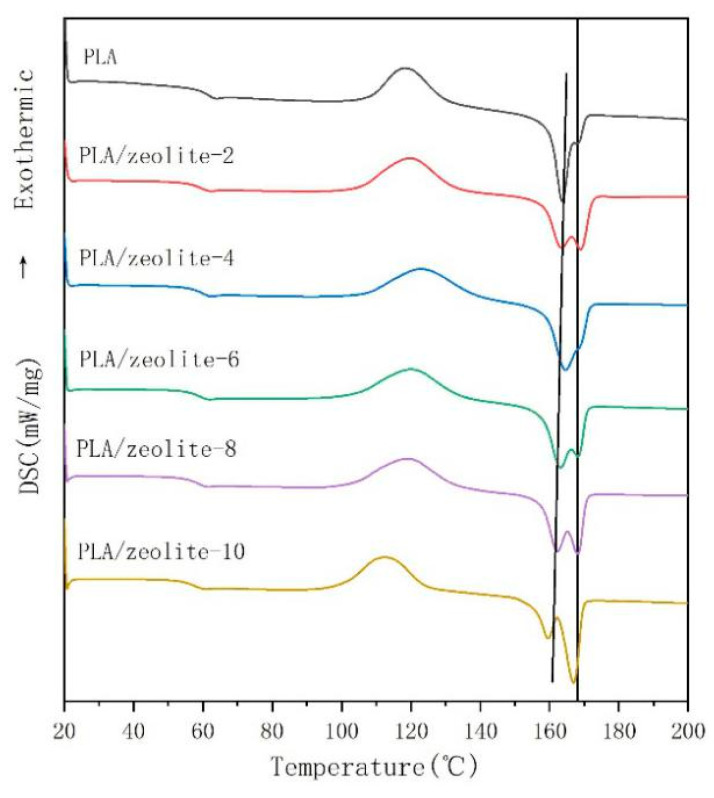
DSC second heating curve of PLA and PLA/zeolite composites.

**Figure 6 membranes-11-00640-f006:**
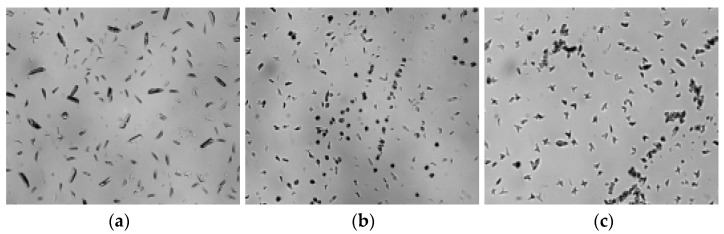
The polarizing microscope photographs of PLA/zeolite-4 (**a**), PLA/zeolite-6 (**b**) and PLA/zeolite-8 (**c**) amplified 100 times.

**Figure 7 membranes-11-00640-f007:**
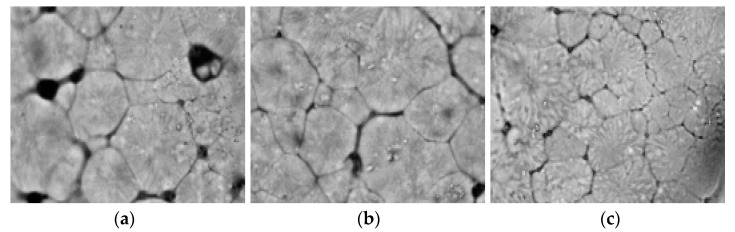
The polarizing microscope photographs of PLA/zeolite-4 (**a**), PLA/zeolite-6 (**b**) and PLA/zeolite-8 (**c**) amplified 1000 times.

**Figure 8 membranes-11-00640-f008:**
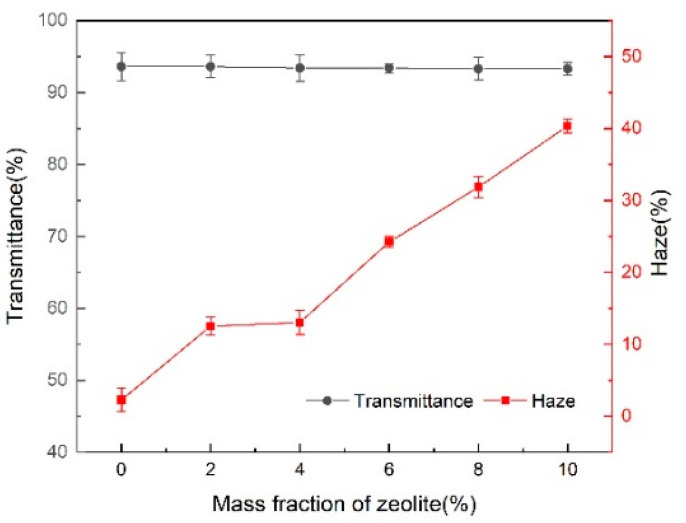
The effect of zeolite on the light transmittance and haze of PLA composites.

**Figure 9 membranes-11-00640-f009:**
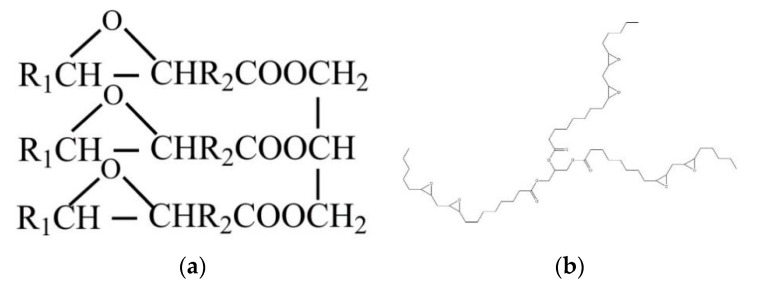
The skeletal structure (**a**) and stereoscopic structure (**b**) of ESO.

**Figure 10 membranes-11-00640-f010:**
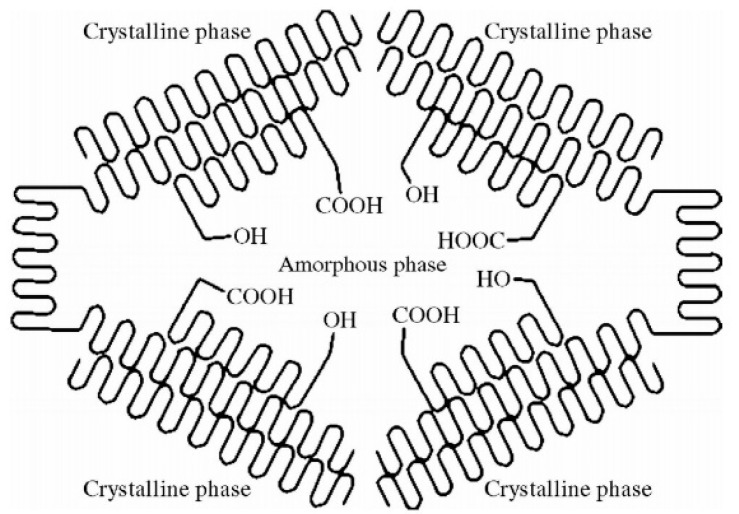
Schematic depiction of solid-state polycondensation.

**Figure 11 membranes-11-00640-f011:**
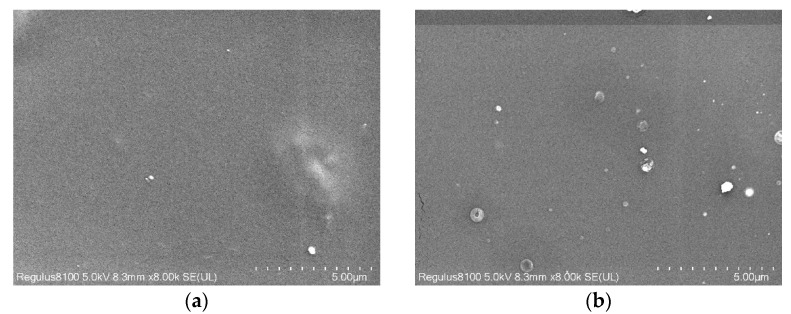
SEM images of PLA (**a**) and PLA/zeolite composites magnified 8000 times with different zeolite concentrations: (**b**–**f**) shows 2, 4, 6, 8 and 10 wt%, respectively.

**Figure 12 membranes-11-00640-f012:**
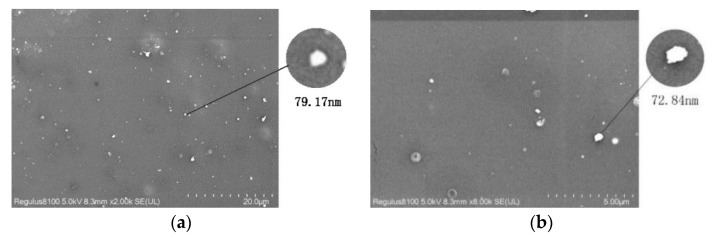
Dispersion size of zeolite in PLA/zeolite-2 magnified 2000 times (**a**) and 8000 times (**b**).

**Figure 13 membranes-11-00640-f013:**
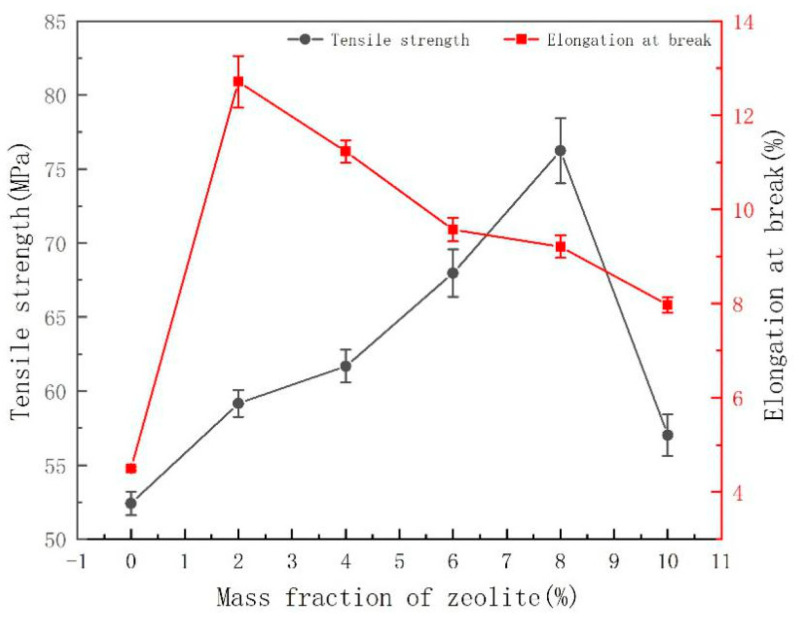
Effect of zeolite mass fraction on the mechanical properties of PLA composite films.

**Table 1 membranes-11-00640-t001:** Experiment recipe.

Samples	Mass Fraction of PLA (wt%)	Mass Fraction of ESO (wt%)	Mass Fraction of Zeolite (wt%)
PLA resin	100	0	0
PLA/zeolite-2	95	3	2
PLA/zeolite-4	93	3	4
PLA/zeolite-6	91	3	6
PLA/zeolite-8	89	3	8
PLA/zeolite-10	87	3	10

**Table 2 membranes-11-00640-t002:** Crystallization and melting parameters of PLA and PLA/zeolite composites.

Samples	Tg (℃)	Tc (℃)	β-Tm (℃)	α-Tm (℃)	∆T (℃)	β-Xc/%	α-Xc/%	Xc/%
PLA	59.50	118.28	163.66	168.11	4.45	28.33	6.30	34.63
PLA/zeolite-2	57.79	119.80	163.41	169.07	5.56	20.21	16.29	36.50
PLA/zeolite-4	58.27	123.10	164.52	168.78	4.26	30.26	10.09	40.35
PLA/zeolite-6	57.41	120.27	163.09	168.11	5.02	24.49	13.97	38.50
PLA/zeolite-8	56.94	119.11	162.28	168.11	5.83	22.87	17.57	40.44
PLA/zeolite-10	56.12	112.95	159.50	166.93	7.43	15.44	28.79	44.24

**Table 3 membranes-11-00640-t003:** The average diameter of spherulites with different zeolite proportions.

Sample	Average Diameter/μm
PLA/zeolite-4	152.48
PLA/zeolite-6	135.74
PLA/zeolite-8	96.57

## Data Availability

Not applicable.

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
