# Peer review of "Effects of Processing Conditions and Plasticizing-Reinforcing Modification on the Crystallization and Physical Properties of PLA Films"

_membranes, 2021, doi:10.3390/membranes11080640_

Round 1
Reviewer 1 Report
The topic of this article is good, however, the article has to be improved significantly in order to publish on Membranes.
- The introduction needs to be rewritten. The current form is just a combination of old references, while there are no references to support a lot of the statements. I recommend the authors can restructure the introduction, use updated references, make sure cite reference to support statements.
- In the experimental section, the manufacturing methods, why the authors chose certain recipes and conditions? The reason should be clearly stated.
- The first and second paragraphs in section 3.1 should not be in the results and discussion section.
- For the images, figures in Figure 10 have scale bars while figures in Figure 4 and 5 were not labeled. The images were not clearly labeled.
- In section 3.3, the authors stated that the use of the green plasticizers and vegetable oil based plasticizers can reduce the price of PLA. How? Any reference to support it? Any data to back it up?
- In the analysis of Figure 10, the authors said "white dots are believed to be zeolite particles" without any characterization methods. No evidence and not convincing; Figure 11 could not use as "calculate of the particle diameter".
- The ultimate reason that I reject this article, is that the authors were not clear what is the significance of the PLA films? The authors did characterize the films, however, did not apply them for any practical or mimetic applications. What is the significance of the PLA films?
- In both abstract and conclusions, the authors did not mention any implications of the results of the synthetic PLA films.
Other than the content defects, the writing style is unprofessional and subjective. Please check written English before resubmitting the manuscript.
Reviewer 2 Report
The work "Effects of processing conditions and plasticizing-reinforcing modification on the crystallization and physical properties of PLA films" presented a relatively well know effects of some additives on polymers processing.
- The authors must present clearly the novelty of the work with a remake introduction (also with some new valuable references).
- The structure and general presentation of manuscript must be improved.
- The images, figures 10 especially, need a high resolution. The presented images show some impurities on the surfaces (a compositional analysis must be realized - EDX).
- The references must wright at journal standards.
Reviewer 3 Report
The studies of the paper were focused on different ratios of zeolite and ESO and their interactions with the crystallization and physical properties of PLA films. The topic is interesting for the readers from the scientific point of view. Authors should add some comments and make changes in the text.
Keywords
- Authors should consider using the addition of „DSC analysis“ to the keywords.
Introduction
- Authors wrote „… Recently, many experts have paid attention to the modification of polylactic acid by physical and chemical methods to overcome its inherent shortcomings, such as brittleness, poor thermal properties and high permeability of gases such as CO2, O2 and water vapour [11]…“ . More papers should be cited (not only one – No. 11).
Experimental
- There is a lack of the number of repetitions or samples used for experiments.
- Authors wrote „… The film was blown to a thickness between 30 μm and 80 μm…“. What was the final thickness of the films?
- What about conditions in the laboratory (temperature, relative humidity)?
- Why the heating rate 10˚C/min has been used for DSC analysis?
- What were the dimensions of the samples for the polarizing microscopy, transparency SEM and strength tests?
Results and Discussion
- Authors should consider remove the information given in 3. Effect of ESO and zeolite addition on the structure and physical properties of PLA films to the previous chapter Experimental.
- Figure 8 is not necessary.
Conclusions
- In the first sentences Authors wrote „… A melting extrusion blow moulding method was used to process a series of PLA/ESO/zeolite blend films. The processing conditions of the extruder were determined at a rotating speed of 35 rpm and temperature T2 (155, 170, 175, 185°C)…“. It is not conclusion and should be deleted.
References
- Authors correctly selected and cited 36 papers.
I recommend the paper for the publishing after minor changes and additions.
Round 2
Reviewer 1 Report
Previously I rejected the manuscript, however, the authors did a phenomenal job editing the manuscript and the manuscript improved significantly. If the authors can have any native speaker go through the paper, the paper will upgrade even more.
Reviewer 2 Report
The remake work "Effects of processing conditions and plasticizing-reinforcing modification on the crystallization and physical properties of PLA films" respond at all gives suggestions.